# A Variational Edge Partition Model for Supervised Graph Representation Learning

**Yilin He**[1*], **Chaojie Wang**[2*†], **Hao Zhang**[3], **Bo Chen**[3], **Mingyuan Zhou**[1†]

[1]The University of Texas at Austin, [2]Nanyang Technological University, [3]Xidian University

{yilin.he,mingyuan.zhou}@mccombs.utexas.edu   chaojie.wang@ntu.edu.sg

zhanghao_xidian@163.com   bchen@mail.xidian.edu.cn

## Abstract

Graph neural networks (GNNs), which propagate the node features through the edges and learn how to transform the aggregated features under label supervision, have achieved great success in supervised feature extraction for both node-level and graph-level classification tasks. However, GNNs typically treat the graph structure as given and ignore how the edges are formed. This paper introduces a graph generative process to model how the observed edges are generated by aggregating the node interactions over a set of overlapping node communities, each of which contributes to the edges via a logical OR mechanism. Based on this generative model, we partition each edge into the summation of multiple community-specific weighted edges and use them to define community-specific GNNs. A variational inference framework is proposed to jointly learn a GNN-based inference network that partitions the edges into different communities, these community-specific GNNs, and a GNN-based predictor that combines community-specific GNNs for the end classification task. Extensive evaluations on real-world graph datasets have verified the effectiveness of the proposed method in learning discriminative representations for both node-level and graph-level classification tasks.

## 1 Introduction

Many real-world entities are bonded by relations, *e.g.*, the users in a social network service are connected by online friendships, and the atoms in molecules are held together by chemical bonds. Representing the entities by nodes and relations by edges, a set of interconnected entities naturally forms a graph. Reasoning about these entities and their relations could be conducted under graph neural networks (GNNs) [1–4]. Instead of isolating the feature transformation for each individual entity, GNNs allow the information to be exchanged between related entities. Models employing this strategy have achieved great success in a wide range of applications involving graph-structured data, such as classification [5, 6], link prediction [7], recommendation [8, 9], (overlapping) community detection [10–13], and drug discovery [14].

The essence of many graph-analytic tasks could be summarized as supervised graph representation learning, where the information on the graph is usually characterized by the node features, adjacency matrix, and node or graph labels. A supervised machine learning pipeline is often introduced to embed the nodes or graph under label supervision for a specific classification task. For most of the GNN-based methods, the information on nodes and edges is composited through neighborhood aggregation, *i.e.*, updating the features of each node by combining the features of the surrounding nodes. During this process, the graph structure mostly serves as a given source to indicate the nodes' neighborhood [15] or provide the weights for aggregation [5]. Treating the graph structure as given

---

*Equal contribution. † Corresponding authors. Code available at `https://github.com/YH-UtMSB/VEPM`

36th Conference on Neural Information Processing Systems (NeurIPS 2022).

overlooks the latent structures that control the formation of the graph, which, however, could provide valuable information for graph representation learning. In other words, failing to consider the latent graph structures may limit the ultimate potential of this line of work.

One type of latent structure at the hinge of the graph structure and node information is node communities. Their connection to the graph can be explained under a latent-community-based graph generation process. For example, both the mixed-membership stochastic blockmodel (MMSB) [10] and edge partition model (EPM) [11] explain the formation of edges by node interactions over overlapping latent communities. Let us consider person $u$ in social network $\mathcal{G}$, some of her social connections may be established through interactions with colleagues as a machine learning researcher, some may originate from her interactions with co-members of the same hobby club, and different types of interactions could overlap to strengthen certain social connections between the nodes.

The node community structure also sets the aspects of node information. In the example of social network $\mathcal{G}$ where person $u$ is affiliated with both a research group and a few hobby groups, due to the diverse nature of these different communities, it is likely that she exhibits different characteristics when interacting with co-members of different communities. Namely, the information on $u$ for the research group may be related to her research expertise, and the information on $u$ for the hobby clubs may be related to her hobbies. Therefore, an ideal solution is to learn a different aspect of node properties for each community that it is affiliated with, and represent the overall node information as an aggregation of all community-specific node properties.

Based on this insight, we develop a *variational edge partition model* (VEPM), which is a generative graph representation learning framework. VEPM models how the edges and labels are generated from overlapping latent communities. Instead of hard assigning a node to a single community, VEPM encodes each node into a $K$-dimensional vector of scores measuring the strength that the node is affiliated with each of the $K$ communities. From these scores, we compute the intensities of pairwise interactions within each community. Given that information, under the Bernoulli-Poisson link, the edges could be modeled by the logical OR of independently generated binary latent edges [11].

The generation of labels includes community-specific node representation learning and aggregation. A key premise of the first part is to detect the hidden structure specified for each community. In our model, it is achieved through edge partition, *i.e.*, decomposing each edge into a summation of link strengths according to the intensities of community-specific interactions. The edge partition step isolates the link strengths accumulated through each community. With the partitioned edges, we learn $K$ separate GNNs to produce community-specific node representations, then compose them into the overall node embeddings to generate the labels. Evaluation shows that the proposed framework can achieve significant performance enhancement on various node and graph classification benchmarks.

We summarize our main contributions as follows:

- We introduce VEPM that utilizes the idea of edge partitioning to extract overlapping latent community structures, which are used to not only enrich node attributes with node-community affiliation scores, but also define community-specific node feature aggregations.
- We formalize the training of VEPM in a variational inference framework, which is powered by GNNs in its latent community inference, representation generation, and label prediction.
- We analyze how VEPM works and evaluate it over various real-world network datasets. Empirical results show that the proposed VEPM outperforms many previous methods in various node- and graph-level classification tasks, especially with limited labels.

## 2   Preliminaries and Related Work

**Embedding nodes and graphs with GNNs.** Given a node-attributed graph $\mathcal{G}$ with $N$ nodes, its information is generally expressed by a design matrix $\mathbf{X} \in \mathbb{R}^{N \times F}$, whose rows represent node features of dimension $F$, and an adjacency matrix $\mathbf{A}$ of shape $N \times N$. The nonzero values of $\mathbf{A}$ indicate the weights on the corresponding edges. GNNs [1–5, 15, 16] are multi-layer parametric models that maps $(\mathbf{X}, \mathbf{A})$ to the embedding space. The update of the embedding of node $u$ at the $t$th GNN layer could be summarized as $\mathbf{h}_u^{(t)} = \mathrm{AGG}(f_{\boldsymbol{\theta}}(\mathbf{h}_u^{(t-1)}), \{f_{\boldsymbol{\theta}}(\mathbf{h}_v^{(t-1)})\}_{v \in \mathbb{N}(u)}, \mathbf{A}_{u,:})$ and $\mathbf{h}_u^{(0)} = \mathbf{x}_u$. Here AGG denotes the neighborhood aggregation function, $\mathbb{N}(u)$ denotes the neighbors of $u$, and $f_{\boldsymbol{\theta}}(\cdot)$ is a transformation function parameterized with $\boldsymbol{\theta}$. In VEPM, we adopt

the aggregation function introduced by Kipf and Welling [5], which can be expressed as $\mathbf{h}_u^{(t)} = \sum_{v \in \{u\} \cup \mathbb{N}(u)} \tilde{A}_{u,v} f_{\boldsymbol{\theta}}(\mathbf{h}_v^{(t-1)})$, where $\tilde{\mathbf{A}}$ is the normalized adjacency matrix of $\mathcal{G}$ augmented with self-loops. When the task requires a graph-level representation, all node embeddings would be summarized into a single embedding vector. This process is called graph pooling [17–19]. In this work, we focus on improving the overall performance from GNN's perspective and adopt a simple graph pooling method proposed by Xu et al. [6]. In the sequel, we unify the expression of models consisting of GNN layers as $\text{GNN}_{\boldsymbol{\theta}}(\mathbf{X}, \mathbf{A})$.

**Community-regularized representation learning.** Communities are latent groups of nodes in a graph [20]. The simplest way to use community information to help a supervised learning task is to regularize it with a community-detection related task. For instance, Hasanzadeh et al. [21] and Wang et al. [22] jointly train a deep stochastic blockmodel and a classification model; a similar approach could be found in Liu et al. [23], where the graph reconstruction regularizer is replaced by a modularity metric. Either way, the node embeddings are expected to reflect patterns of community structures and maintain informative to the task at the same time. These methods generally treat node-community affiliation as node embeddings and expect them to contain sufficient information for the downstream task. However, the affiliation strengths of nodes to a community may oversimplify the information that such a community can provide for the downstream task. Unlike these methods, VEPM embeds node information on each community with a specific GNN, where the community structures are reflected by the partitioned adjacency matrices. Aggregating node features with these weighted adjacency matrices naturally incorporates community information into task-learning.

**Multi-relational data analysis.** Leveraging heterogeneous relations has been extensively studied in the literature of mining Knowledge Graphs [24–26]. The data of these models is usually organized by a multigraph, which is related to our perspective of the latent node interactions that generate the observed graph because the interactions in different communities are potentially heterogeneous, and they are permitted to coexist between a pair of nodes. This leads to a shared high-level idea to break down the original heterogeneous graph into homogeneous factors and analyze each factor with a customized model. However, unlike those multigraphs where the edge types are explicitly annotated, the overlapping heterogeneous node interactions are not observable from the data that VEPM aims to handle, escalating the difficulty of our optimization problem. To deal with the latent variables, we formalize VEPM into a generative model and train it via variational inference.

**Graph factorization-based models.** The existence of multiple hidden structures in graphs is also noticed by some previous work [27, 28]. The architecture of these models starts with graph factorization, followed by a set of modules with each one processing the information from one of the graph factors. In these methods, the only ground-truth information that trains graph factorization is from label supervision, making them rely on excessive label annotations, which might be hard to suffice in practice and thus potentially hinder their application. On account of the potential heterogeneity of graph information on different communities, VEPM adopts a similar pipeline to these methods, but the factorization of the original graph is driven by not only the supervised task but also a graph generative model, which greatly reduces our dependency on the amount of labeled data.

# 3 Variational Edge Partition Model

Many analytical tasks on node-attributed graphs could boil down to (semi-) supervised classification problems, *i.e.*, given $(\mathbf{X}, \mathbf{A})$ and observed node or graph labels $\mathbf{y}_o$, predicting the unobserved labels $\mathbf{y}_u$. In VEPM, we formalize the classification task as modeling the predictive distribution as $p(\mathbf{y}_u \mid \mathbf{X}, \mathbf{A}, \mathbf{y}_o)$, and construct our variational supervised learning pipeline with GNNs.

## 3.1 Generative task-dependent graph representation learning with latent communities

To model $p(\mathbf{y}_u \mid \mathbf{X}, \mathbf{A}, \mathbf{y}_o)$, we introduce $\mathbf{Z} \in \mathbb{R}_+^{N \times K}$, a latent non-negative node-community affiliation matrix, whose entry at the $n$th row and $k$th column is interpreted as the strength that the $n$th node is affiliated with the $k$th community. Given $\mathbf{Z}$, we specify a *generative network* as

$$\mathbf{A} \sim p_{\boldsymbol{\theta}}(\mathbf{A} \mid \mathbf{Z}), \;\; \mathbf{y}_o \sim p_{\boldsymbol{\theta}}(\mathbf{y} \mid \mathbf{X}, \mathbf{A}, \mathbf{Z}) \qquad (1)$$

to describe how the observed edges $\mathbf{A}$ and labels $\mathbf{y}_o$ are generated. Due to the complexity brought by the deep architecture of the *generative network*, analytically inferring the posterior of $\mathbf{Z}$ is impractical.

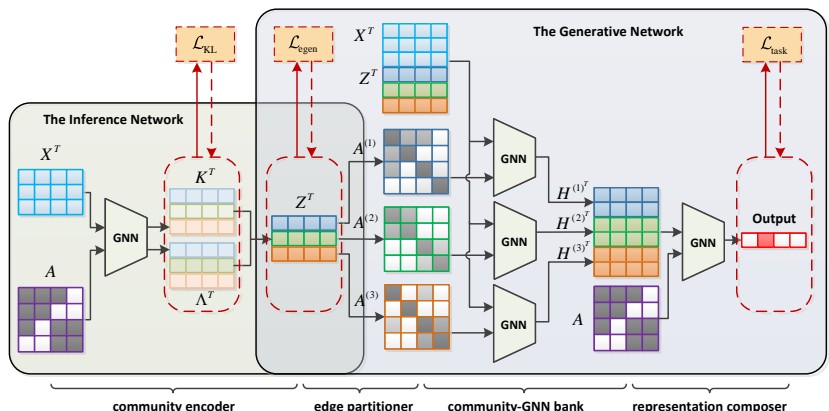

Figure 1: The overview of VEPM's computation graph. The input data successively passes four modules described in Sections 3.2 and 3.3: the *community encoder*, *edge partitioner*, *community-GNN bank*, and *representation composer*. Node information is processed community-wise by the *edge partitioner* and the *community-GNN bank*. The former factorizes the original graph into a set of weighted graphs by the relative inter-node interaction rates across communities; the latter performs neighborhood aggregation with the factorized graphs, such that in each community, nodes exchange more information with intensively interacted neighbors. Hence the obtained node representations are community-specific. Please note that we use the transposed notation for $\mathbf{Z}$ and node embeddings in this diagram, so the concatenations are along the vertical direction. Solid and dashed arrows indicate network forwarding and gradient back-propagation, respectively.

We hence approximate the posterior $p_{\boldsymbol{\theta}}(\mathbf{Z} \mid \mathbf{A}, \mathbf{y}_o)$ with a variational distribution, $q_{\boldsymbol{\phi}}(\mathbf{Z} \mid \mathbf{A}, \mathbf{X})$, modeled by a separate GNN-based *inference network*. The subscripts $_{\boldsymbol{\theta}}$ and $_{\boldsymbol{\phi}}$ denote the parameters in the *generative network* and *inference network*, respectively. We illustrate the architecture of VEPM in Figure 1. We approximate the posterior predictive distribution using the Monte Carlo method as

$$\hat{p}_{\boldsymbol{\theta}}(\mathbf{y}_u \mid \mathbf{X}, \mathbf{A}, \mathbf{y}_o) \approx \tfrac{1}{S} \sum_{s=1}^{S} p_{\boldsymbol{\theta}}(\mathbf{y}_u \mid \mathbf{X}, \mathbf{A}, \mathbf{Z}^{(s)}), \tag{2}$$

where $\mathbf{Z}^{(s)} \overset{iid}{\sim} q_{\boldsymbol{\phi}}(\mathbf{Z} \mid \mathbf{A}, \mathbf{X})$ for $s = 1, \ldots, S$.

In what follows, we outline the data generation process and corresponding modules in the *generative network*, introduce the module in the *inference network*, and describe how to train both networks.

### 3.2 Edge generation and label prediction

To model the distribution of the edges given $\mathbf{Z}$, we adopt the generative process developed in EPM [11], which explains the generation of the edges under the Bernoulli-Poisson link as $\mathrm{A}_{i,j} = \mathbf{1}_{\mathrm{M}_{i,j} \geq 1}$, $\mathrm{M}_{i,j} \sim \mathrm{Poisson}\left(\sum_{k=1}^{K} \gamma_k \mathrm{Z}_{i,k} \mathrm{Z}_{j,k}\right)$. Here $\gamma_k$ is a positive community activation level indicator, which measures the member interaction frequency via community $k$, and in our practice is treated as a trainable parameter shared by both the *inference network* and *generative network*. We interpret $\gamma_k \mathrm{Z}_{i,k} \mathrm{Z}_{j,k}$ as the interaction rate between nodes $i$ and $j$ via community $k$.

The EPM has an alternative representation that partitions each edge into the logical disjunction (*i.e.*, logical OR) of $K$ latent binary edges [29], expressed as

$$\mathrm{A}_{i,j} = \vee_{k=1}^{K} \mathrm{A}_{i,j,k}, \ \mathrm{A}_{i,j,k} \sim \mathrm{Bernoulli}(p_{ijk}),$$

where $p_{ijk} := 1 - e^{-\gamma_k \mathrm{Z}_{i,k} \mathrm{Z}_{j,k}}$. Thus nodes $i$ and $j$ would be connected as long as their interaction forms an edge in at least one community. In other words, they are disconnected only if their interactions in all communities fail to generate any edges. Under the EPM, we have the conditional probability as $p_{\boldsymbol{\theta}}(\mathrm{A}_{i,j} = 1 \mid \mathbf{Z}) = 1 - e^{-\sum_{k=1}^{K} \gamma_k \mathrm{Z}_{i,k} \mathrm{Z}_{j,k}}$. To complete the model, we set the prior distribution of the node-community affiliation matrix as $\mathbf{Z} \sim \prod_{i=1}^{N} \prod_{k=1}^{K} \mathrm{Gamma}(\mathrm{Z}_{i,k}; \alpha, \beta)$.

Given $\mathbf{A}$ and $\mathbf{Z}$, we now describe the predictive process of the labels. The node-community affiliation impacts this process from two aspects: for each node, the corresponding row of $\mathbf{Z}$ serves as side information that could enrich node attributes; for each pair of connected nodes, it could be used to derive the node interactions rate via each community, *i.e.*, $\{\gamma_k \mathrm{Z}_{i,k} \mathrm{Z}_{j,k}\}_{k=1,K}$, which are further used to partition the edges into $K$ communities, carried out by the *edge partitioner*:

**Edge partitioner.** *The edge partitioner takes edges* $\mathbf{A}$ *and node-community affiliation matrix* $\mathbf{Z}$ *as inputs and returns* $K$ *positive-weighted edges:* $\{\mathbf{A}^{(1)}, \mathbf{A}^{(2)}, \cdots, \mathbf{A}^{(K)}\}$, s.t. $\sum_{k=1}^{K} \mathbf{A}^{(k)} = \mathbf{A}$. *The edge partition function is*

$$\mathrm{A}_{i,j}^{(k)} = \mathrm{A}_{i,j} \cdot \frac{e^{(\gamma_k \mathrm{Z}_{i,k} \mathrm{Z}_{j,k})/\tau}}{\sum_{k'} e^{(\gamma_{k'} \mathrm{Z}_{i,k'} \mathrm{Z}_{j,k'})/\tau}}, \ k \in \{1, \ldots, K\}, \ i,j \in \{1, \ldots, N\}, \tag{3}$$

*where* $\tau$ *is a "temperature" that controls the sharpness of partition.*

The effect of the *edge partitioner* could be considered as a soft assignment of each edge into different communities. Setting temperature $\tau$ to be low could drive the soft assignment towards a hard assignment, and consequently, when aggregating node features with $\{\mathbf{A}^{(1)}, \mathbf{A}^{(2)}, \cdots, \mathbf{A}^{(K)}\}$, the edge partitioner would concentrate the information exchange between any two connected nodes at one community. In other words, the mutual influence between two nodes, measured by the edge weight, is relatively high in the community that contributes the most of the interactions between the pair, while being relatively weak in the other communities.

In our implementation, we further generalize Equation (3) to a metacommunity-based edge partition, specifically, by replacing $\gamma_k \mathrm{Z}_{i,k} \mathrm{Z}_{j,k}$ with $\mathbf{Z}_{i,k} \mathrm{diag}(\boldsymbol{\gamma}_k) \mathbf{Z}'_{j,k}$. In the new expression, $\mathbf{Z}_{i,k}$ denotes the $k$th segment in node $i$'s community-affiliation encoding, $\boldsymbol{\gamma}_k$ is a vector of the activation levels for communities in the $k$th metacommunity. This generalization allows the total number of communities to be greater than $K$, enhancing the model implementation flexibility at a minor interpretability cost.

The *edge partitioner* provides a soft separation of the neighborhood aggregation routes for each community, which is passed to the *community-GNN bank*:

**Community-GNN bank.** *The module takes* $\mathbf{X}^* := \mathbf{X} \parallel \mathbf{Z}$ *as input, where the operator* $\parallel$ *denotes "concatenation," and learns community-specific node embeddings with* $K$ *separate GNNs, i.e., the outputs are* $\mathrm{g}_{\boldsymbol{\theta}}^{(1)}(\mathbf{X}^*), \mathrm{g}_{\boldsymbol{\theta}}^{(2)}(\mathbf{X}^*), \ \cdots, \ \mathrm{g}_{\boldsymbol{\theta}}^{(K)}(\mathbf{X}^*)$, *where* $\mathrm{g}_{\boldsymbol{\theta}}^{(k)}(\cdot) := \mathrm{GNN}_{\boldsymbol{\theta}}(\cdot, \mathbf{A}^{(k)})$, $k \in \{1, \ldots, K\}$.

The intention of the *edge partitioner* and *community-GNN bank* is to capture the node information specific to each community. For instance, in a social network, such kind of information could be the different social roles of people when considered affiliated with different social groups, *e.g.*, one may be a former computer science major student in an alumni group, a research scientist in a work group, and an amateur chess player in a hobby club group. As shown in Section 4.3, the node embeddings produced by the *community-GNN bank* are separable by communities.

We finally introduce the *representation composer*, which embeds a combination of the information provided by each community into a global representation via a GNN:

**Representation composer.** *Let* $\mathbf{H}^{(k)} := \mathrm{g}_{\boldsymbol{\theta}}^{(k)}(\mathbf{X}^*)$ *denote the node representations learned from the* $k$th *community, where* $k = 1, \ldots, K$, *and* $f(\cdot)$ *denote the representation composer, whose functionality is to project a composite of community-specific node representations to one representation matrix, i.e.,* $\mathbf{H}_{\mathcal{V}} = f\left(\mathbf{H}^{(1)}, \mathbf{H}^{(2)}, \cdots, \mathbf{H}^{(K)}\right) := \mathrm{GNN}_{\boldsymbol{\theta}}\left(\parallel_{k=1}^{K} \mathbf{H}^{(k)}, \mathbf{A}\right).$

For graph-level tasks, we further pool node embeddings into a single vector representation $\mathbf{h}_{\mathcal{G}}$, as in Xu et al. [6]. Taking softmax on the feature dimension of $\mathbf{H}_{\mathcal{V}}$ or $\mathbf{h}_{\mathcal{G}}$ gives the predicted probabilities of labels, from which we are able to classify the unlabeled objects. Cascading the *edge partitioner* and *community-GNN bank* with the *representation composer* yields the probability $p_{\boldsymbol{\theta}}(\mathbf{y} \mid \mathbf{X}, \mathbf{A}, \mathbf{Z})$.

### 3.3 Variational latent community inference

We use a *community encoder* as the module in the *inference network* to approximate the posterior distribution $p_{\boldsymbol{\theta}}(\mathbf{Z} \mid \mathbf{A}, \mathbf{y})$, and provide the *generative network* with the critical latent variable $\mathbf{Z}$:

**Community encoder.** *The community encoder models the variational posterior* $q_{\boldsymbol{\phi}}(\mathbf{Z} \mid \mathbf{A}, \mathbf{X})$ *by a Weibull distribution with shape* $\mathbf{K}$ *and scale* $\boldsymbol{\Lambda}$, *whose parameters are learned by a GNN as* $\mathbf{K} \parallel \boldsymbol{\Lambda} = \mathrm{GNN}_{\boldsymbol{\phi}}(\mathbf{X}, \mathbf{A})$. *A Weibull random sample from* $q_{\boldsymbol{\phi}}(\mathbf{Z} \mid \mathbf{A}, \mathbf{X})$ *could be created through the inverse CDF transformation of a uniform random variable, given as follows:*

$$\mathbf{Z} = \boldsymbol{\Lambda} \odot \left(-\log(1 - \mathbf{U})\right)^{(1 \oslash \mathbf{K})}, \mathrm{U}_{i,k} \overset{iid}{\sim} \mathrm{U}(0,1), \ \forall (i,k) \in \{1, \ldots, N\} \times \{1, \ldots, K\}, \tag{4}$$

*where* $\odot$ *and* $\oslash$ *denote element-wise multiplication and division.*

## 3.4 The overall training algorithm and complexity analysis

We train VEPM by optimizing the evidence lower bound (ELBO), decomposed into three terms, as

$$\mathcal{L} = \mathcal{L}_{\text{task}} + \mathcal{L}_{\text{egen}} + \mathcal{L}_{\text{KL}}, \tag{5}$$

where $\mathcal{L}_{\text{task}} = \mathbb{E}_{q_{\phi}(\mathbf{Z}\,|\,\mathbf{A},\mathbf{X})} \log p_{\theta}(\mathbf{y}_o \,|\, \mathbf{A}, \mathbf{Z}, \mathbf{X})$, $\mathcal{L}_{\text{egen}} = \mathbb{E}_{q_{\phi}(\mathbf{Z}\,|\,\mathbf{A},\mathbf{X})} \log p_{\theta}(\mathbf{A}\,|\,\mathbf{Z})$, and $\mathcal{L}_{\text{KL}} = -D_{\text{KL}}\big(q_{\phi}(\mathbf{Z}\,|\,\mathbf{A},\mathbf{X}) \,\|\, p(\mathbf{Z})\big)$. These three terms correspond to the classification task, edge generation, and KL-regularization, respectively. Note that our specifications of $\mathbf{Z}$'s prior and variational posterior yield an analytical expression of $\mathcal{L}_{\text{KL}}$, as described in detail in Appendix B.

Recall that $N$, $M$, $K$ denote the numbers of nodes, edges, and communities (or metacommunities) in the graph; $F$ denotes the feature dimension; and $L_1, L_2, L_3$ denote the number of layers in the *community encoder*, in the *community-GNN bank*, and in the *representation composer*. In VEPM, we limit the size of hidden dimension in each *community-GNN* to $1/K$ of what is commonly used among the baselines, hence the time complexity of training VEPM is $\mathcal{O}((L_1 + L_2 + L_3)MF + (L_1 + L_2/K + L_3)NF^2 + N^2F)$. For a sparse graph where $N^2 \gg M$, the computational overhead can be reduced to $\mathcal{O}(M)$ [2] if graph reconstruction is accelerated via subsampling the nodes to $\mathcal{O}(\sqrt{M})$, as in Salha et al. [30]. Effects and implications of adopting such type of acceleration algorithms are discussed in Appendix A. The space complexity of VEPM is $\mathcal{O}((L_1+L_2+L_3)NF+KM+(L_1+L_2/K+L_3)F^2)$, among which $\mathcal{O}((L_1 + L_2/K + L_3)F^2)$ is contributed by model parameters. It is noteworthy to point out that the memory cost of graph reconstruction is manageable by computing the dense matrix multiplication block-wise with fixed maximum block size. From all aspects of complexity, VEPM (with acceleration) is comparable to GATs [15, 31] and models involving graph factorization [27, 28].

# 4 Empirical Evaluation

## 4.1 Node & graph classification

**Datasets & experimental settings.** For node classification, we consider three citation networks (Cora, Citeseer, and Pubmed) and a Wikipedia-based online article network (WikiCS) [32], which provide either bag-of-words document representations or average word embeddings as node features, and (undirected) citations or hyperlinks as edges; For graph classification, we consider four

Table 1: Comparison of node classification performance.

| Method | Cora | Citeseer | Pubmed | WikiCS |
|---|---|---|---|---|
| ChebNet [4] | 81.2 | 69.8 | 74.4 | - |
| GCN [5] | 81.5 | 70.3 | 79.0 | $74.0 \pm 1.0$ |
| GCN-64 | 81.4 | 70.9 | 79.0 | - |
| SIG-VAE [21] | 79.7 | 70.4 | 79.3 | - |
| WGCAE [22] | 82.0 | 72.1 | 79.1 | - |
| GAT* [15] | 83.0 | 72.5 | 79.0 | $77.6 \pm 0.6$ |
| hGANet [31] | 83.5 | 72.7 | 79.2 | - |
| DisenGCN [27] | 83.7 | **73.4** | 80.5 | - |
| VEPM (this work) | **$84.3 \pm 0.1$** | $72.5 \pm 0.1$ | **$82.4 \pm 0.2$** | **$78.7 \pm 0.6$** |

bioinformatics datasets (MUTAG, PTC, NCI1, PROTEINS) and four social network datasets (IMDB-BINARY, IMDB-MULTI, REDDIT-BINARY, REDDIT-MULTI). The input node features are crafted in the same way as Xu et al. [6]. Most of the baselines are compared following the 10-fold cross-validation-based evaluation protocol proposed by Xu et al. [6]. For the graph classification task, we also evaluate our model following Zhang and Chen [33], which conducts a more rigorous train-validation-test protocol. More details about the experiments are elaborated in Appendix C.2.

**Node classification.** We use classification accuracy as the evaluation metric for node classification. Table 1 reports the average performance of VEPM ($\pm$ standard error) against related baselines that are categorized into three different groups. The first group consists of the GCNs [5] and its variant GCN-64, which expands the hidden dimension from 16 to 64. VEPM outperforms the first group by a significant margin. The explanation is twofold: (i) VEPM augments the node attributes with community information carried by the inferred node-community affiliations, (ii) VEPM learns community-specific node embeddings. The second group includes SIG-VAE [21] and WGCAE [22], both of which learn node embeddings from graph generative models jointly optimized with a supervised loss. The performance gain that VEPM obtains could be attributed to our unique label predictive process, which not only appends extra community information to node attributes but also injects structural patterns of communities into neighborhood aggregation. The third group [15, 31, 27] is focused on learning node embeddings leveraging heterogeneous hidden relations. When fitting the hidden relations via the attention mechanism, they only use information from the node features

---

[2] For expression simplicity, the scale of $F$ is treated as constant, which is ubiquitous in practice.

Table 2: Comparison of graph classification performance (average accuracy ± standard error).

| Method | IMDB-B | IMDB-M | MUTAG | PTC | NCI1 | PROTEINS | RDT-B | RDT-M |
|---|---|---|---|---|---|---|---|---|
| WL subtree [34] | 73.8 ± 3.9 | 50.9 ± 3.8 | 90.4 ± 5.7 | 59.9 ± 4.3 | **86.0** ± 1.8 | 75.0 ± 3.1 | 81.0 ± 3.1 | 52.5 ± 2.1 |
| PATCHYSAN [35] | 71.0 ± 2.2 | 45.2 ± 2.8 | 92.6 ± 4.2 | 60.0 ± 4.8 | 78.6 ± 1.9 | 75.9 ± 2.8 | 86.3 ± 1.6 | 49.1 ± 0.7 |
| AWE [36] | 74.5 ± 5.9 | 51.5 ± 3.6 | 87.9 ± 9.8 | - | - | - | 87.9 ± 2.5 | 54.7 ± 2.9 |
| GAT [15] | 70.5 ± 2.3 | 47.8 ± 3.1 | 89.4 ± 6.1 | 66.7 ± 5.1 | 60.8 ± 2.5 | 70.5 ± 3.6 | - | 44.4 ± 2.6 |
| hGANet [31] | - | 49.0 | 90.0 | 65.0 | - | 78.7 | - | - |
| GIN [6] | 75.1 ± 5.1 | 52.3 ± 2.8 | 89.4 ± 5.6 | 64.6 ± 7.0 | 82.7 ± 1.6 | 76.2 ± 2.8 | **92.4** ± 2.5 | **57.5** ± 1.5 |
| R-GCN [25] | - | - | 82.3 ± 9.2 | 67.8 ± 13.2 | - | - | - | - |
| CompGCN [26] | - | - | 89.0 ± 11.1 | 71.6 ± 12.0 | - | - | - | - |
| FactorGCN [28] | 75.3 ± 2.7 | 51.0 ± 3.1 | 89.9 ± 6.5 | 74.3 ± 8.9 | 76.4 ± 2.1 | 74.2 ± 4.6 | 90.0 ± 1.2 | 46.5 ± 2.1 |
| VEPM (this work) | **76.7** ± 3.1 | **54.1** ± 2.1 | **93.6** ± 3.4 | **75.6** ± 5.9 | 83.9 ± 1.8 | **80.5** ± 2.8 | 90.5 ± 1.8 | 55.0 ± 1.5 |

Table 3: Graph classification accuracies under the protocol by Zhang and Chen [33].

| Method | IMDB-B | IMDB-M | MUTAG | NCI1 | PROTEINS | RDT-M |
|---|---|---|---|---|---|---|
| WL subtree [34] | 73.4 ±4.6 | 49.3 ±4.8 | 82.1 ±0.4 | 82.2 ±0.2 | 74.7 ±0.5 | 49.4 ±2.4 |
| DGCNN [37] | 70.0 ±0.9 | 47.8 ±0.9 | 85.8 ±1.7 | 74.4 ±0.5 | 75.5 ±0.9 | 48.7 ±4.5 |
| GCAPS-CNN [38] | 71.7 ±3.4 | 48.5 ±4.1 | - | **82.7** ±2.4 | 76.4 ±4.2 | 50.1 ±1.7 |
| CapsGNN [33] | 73.1 ±4.8 | 50.3 ±2.7 | 86.7 ±6.9 | 78.4 ±1.6 | 76.3 ±3.6 | **52.9** ±1.5 |
| VEPM (this work) | **74.6** ±2.1 | **50.6** ±3.6 | **91.5** ± 4.3 | 81.7 ± 1.4 | **76.8** ± 4.1 | 50.2 ± 2.1 |

through label supervision, whereas our approach also takes the observed graph structure into account via a graph generative model. The additional information from the graph utilized by VEPM could explain the enhancement achieved by VEPM over the third group.

**Graph classification.** For the first protocol, we compare VEPM with classical graph classification baselines [34–36], generic-GNN-based models [15, 31, 6], and GNNs with relation-based or task-driven graph factorization [25, 26, 28]. Results in Table 2 show that VEPM achieves the best graph classification performance on 5 out of 8 benchmarks, and the second-best performance on the other 3 benchmarks, including NCI1, where VEPM outperforms all the other GNN-based models.

For the second protocol, aside from a traditional method, WL [34], and a generic-GNN-based method, DGCNN [37], we compare VEPM with two GNNs [38, 33] that adapt capsule neural networks [39] to graph-structured data, which aim to learn different aspects of graph properties via dynamic routing [40]. The results in Table 3 show that VEPM outperforms these GNN-based methods on most of the benchmarks. This indicates that the aspects of graph information, as defined with communities and learned via aggregating node features with partitioned graphs, are more pertinent to the end tasks.

**Classification with reduced labels.** The gain of incorporating the graph generation process into VEPM is that the observed graph structure also provides information for hidden community detection and edge partition, which is beneficial if the amount of labeled data is not sufficient for both decomposing the graph into hidden factors and semi-supervised task-learning. To illustrate this point, based on the evaluation protocol by Kipf and Welling [5] for node classification and Xu et al. [6] for graph

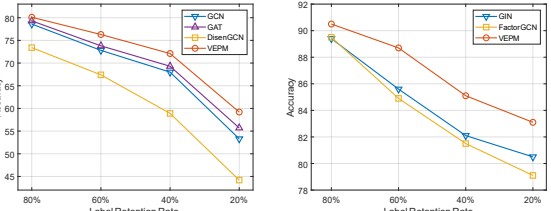

Figure 2: Node classification accuracy on Cora (left) and graph classification accuracy on MUTAG (right) as the number of labels decreases. The horizontal axis is the keep rate of labels in the training set under corresponding protocols.

classification, we re-evaluate the performance of VEPM under reduced training labels on Cora (for node classification) and MUTAG (for graph classification), along with generic-GNN-based methods [5, 15, 6] and decomposition-based methods [27, 28]. The results are shown in Figure 2: the performance of all models is negatively impacted by the reduction in training labels, those suffering from the strongest performance decline are DisenGCN [27] and FactorGCN [28] that decompose graphs solely based on label supervision, while VEPM also performs graph factorization with limited labels, it consistently outperforms the other methods under all settings, and the superiority becomes even more evident as the amount of annotated data further decreases. We consider the enhanced robustness under sparsely labeled data as a crucial feature to have for real-world graphs.

## 4.2 Ablation Studies

**Different edge partition schemes.** In this part, we study how much a meaningful edge partition would benefit the task. Except for the partition we obtained from training the regular VEPM, we tested the performance of VEPM under two edge partition schemes: even partition and random partition. Even partition means all edge weights are $1/K$. For random partition, we sample the unnormalized

edge weights from $U(0, 100)$, then normalize them along the community dimension with `softmax`. The random edge weights are fixed for training to simulate an undesirable convergence state of edge partition. The results are recorded in Table 4. Generally speaking, the performance of VEPM with even edge partition is comparable to that of GIN, our base model; when we substitute random partition for even partition, the model performance becomes worse than the base model. These ablation study results suggest that VEPM can benefit from enhancing the quality of edge partitions.

Table 4: The performance of VEPM with different edge partition schemes.

| Method | IMDB-B | IMDB-M | MUTAG | PTC | PROTEINS |
|---|---|---|---|---|---|
| VEPM (even partition) | $74.2 \pm 4.3$ | $51.4 \pm 2.7$ | $90.4 \pm 5.4$ | $66.3 \pm 6.6$ | $76.3 \pm 3.2$ |
| VEPM (random partition) | $60.5 \pm 4.7$ | $41.1 \pm 1.5$ | $81.4 \pm 5.6$ | $61.8 \pm 4.0$ | $62.7 \pm 3.8$ |
| GIN | $75.1 \pm 5.1$ | $52.3 \pm 2.8$ | $89.4 \pm 5.6$ | $64.6 \pm 7.0$ | $76.2 \pm 2.8$ |
| VEPM | $76.7 \pm 3.1$ | $54.1 \pm 2.1$ | $93.6 \pm 3.4$ | $75.6 \pm 5.9$ | $80.5 \pm 2.8$ |

**Different *representation composer*.** Next, we use the partition obtained from a regular VEPM to represent partitions that are meaningful for the task, and a random partition to represent "meaningless partitions." For each partition scheme, we contrast the performance of VEPM between two designs of the *representation composer*: (i) GNN-based design, represented by the current design; (ii) parsimonious design, represented by a fully-connected (FC) layer. The results are in Table 5. When the edge partition is meaningful, *i.e.*, relevant to the task, the overall performance with a simpler *representation composer* is generally slightly worse than the current design, but still better than most of the baselines. However, if the partition is irrelevant to the task, the GNN-based *representation composer* would be the last thing standing between the model and a failed task, as the graph structure used for neighborhood aggregation is still informative to the task. So the current design would slightly benefit VEPM in a general case and protect the performance of the model in the worst-case scenario.

Table 5: The performance of VEPM with different designs of the *representation composer*.

| | *representation composer* design | IMDB-B | IMDB-M | MUTAG | PTC | PROTEINS |
|---|---|---|---|---|---|---|
| meaningful partition | GNN-based | $76.7 \pm 3.1$ | $54.1 \pm 2.1$ | $93.6 \pm 3.4$ | $75.6 \pm 5.9$ | $80.5 \pm 2.8$ |
| | FC-based | $74.6 \pm 3.4$ | $51.6 \pm 1.7$ | $90.3 \pm 4.2$ | $68.7 \pm 4.3$ | $76.7 \pm 3.2$ |
| random partition | GNN-based | $60.5 \pm 4.7$ | $41.1 \pm 1.5$ | $81.4 \pm 5.6$ | $61.8 \pm 4.0$ | $62.7 \pm 3.8$ |
| | FC-based | $58.6 \pm 3.8$ | $45.3 \pm 2.0$ | $78.6 \pm 5.9$ | $58.5 \pm 5.9$ | $59.8 \pm 3.9$ |

**Different edge partitioning temperatures.** We use the same way as the previous study to get meaningful and random partitions. In this study, we focus on the effect of different selections of $\tau$, the temperature parameter of `softmax`. The value of $\tau$ controls the sharpness of the partitioned edge weights, *i.e.*, a small $\tau$ drives the partitioned weights for the same edge towards a one-hot vector, whereas a large $\tau$ would eliminate the divergence among the edge weights and make the edge partition no more different from an even partition. The results of this experiment are recorded in Table 6, the upper half of which is obtained from a meaningful edge partition, and the lower half is obtained from a random edge partition. We can induce from these results that when the basis to perform edge partition, namely, the inferred latent node interactions, is relevant to the task, a sharp edge partition that highlights the differences among the communities may be helpful for the task. Otherwise, a smooth edge partition might be more favorable.

Table 6: The performance of VEPM with different selections of $\tau$.

| | | $\tau = 0.1$ | $\tau = 1$ | $\tau = 10$ | $\tau = 100$ | $\tau = 1000$ |
|---|---|---|---|---|---|---|
| meaningful partition | **MUTAG** | $93.6 \pm 4.3$ | $93.6 \pm 3.4$ | $91.6 \pm 7.6$ | $92.6 \pm 6.3$ | $90.8 \pm 5.2$ |
| | **PTC** | $75.9 \pm 5.5$ | $75.6 \pm 5.9$ | $74.7 \pm 7.8$ | $66.4 \pm 6.8$ | $69.6 \pm 6.4$ |
| | **PROTEINS** | $78.5 \pm 2.6$ | $80.5 \pm 2.8$ | $79.2 \pm 2.4$ | $77.4 \pm 3.9$ | $76.3 \pm 2.7$ |
| random partition | **MUTAG** | $79.2 \pm 7.9$ | $81.4 \pm 5.6$ | $84.5 \pm 6.8$ | $87.7 \pm 5.4$ | $89.8 \pm 4.6$ |
| | **PTC** | $60.1 \pm 4.8$ | $61.8 \pm 4.0$ | $62.5 \pm 5.4$ | $64.6 \pm 4.5$ | $65.2 \pm 4.9$ |
| | **PROTEINS** | $64.8 \pm 4.9$ | $62.7 \pm 3.8$ | $68.7 \pm 2.7$ | $71.7 \pm 3.9$ | $73.3 \pm 2.7$ |

## 4.3 Qualitative analysis

**Visualizing community structures.** To show that through partitioning the edges, VEPM identifies each latent community from the original graph, in Figure 3, we plot the adjacency matrices of a 200-node subgraph of Cora, before and after edge partition. The subgraph is created via breadth-first search [41] node selection to ensure connectivity. We sort the nodes sample $\mathbb{S}$ in order to present a clearer view of the community structures. Specifically, we prepare $K$ buckets; for node $u$, we compute the total interactions it engages under metacommunity $k$ by $\mu_{u,k} := \sum_{v \in \mathbb{S}} \mathbf{Z}_{u,k} \mathrm{diag}(\boldsymbol{\gamma}_k) \mathbf{Z}'_{v,k}$, and assign it to bucket $k'$, where $k' = \arg\max_k \mu_{u,k}$. We first sort the buckets by descending their counts

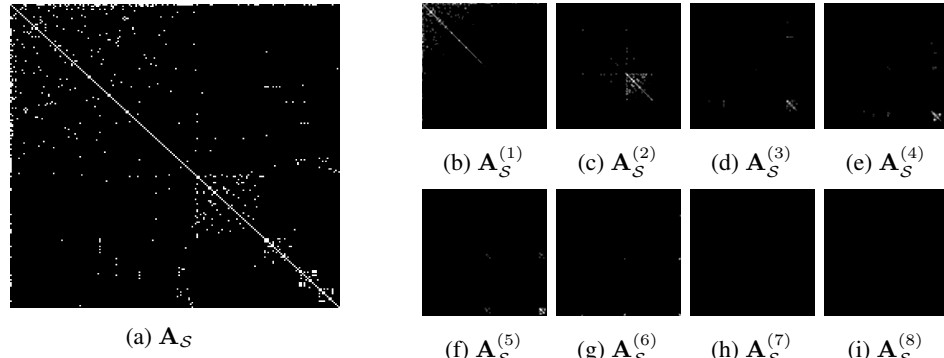

(a) $\mathbf{A}_{\mathcal{S}}$ (b) $\mathbf{A}_{\mathcal{S}}^{(1)}$ (c) $\mathbf{A}_{\mathcal{S}}^{(2)}$ (d) $\mathbf{A}_{\mathcal{S}}^{(3)}$ (e) $\mathbf{A}_{\mathcal{S}}^{(4)}$

(f) $\mathbf{A}_{\mathcal{S}}^{(5)}$ (g) $\mathbf{A}_{\mathcal{S}}^{(6)}$ (h) $\mathbf{A}_{\mathcal{S}}^{(7)}$ (i) $\mathbf{A}_{\mathcal{S}}^{(8)}$

Figure 3: $S$ is a connected subgraph of 200 nodes sampled from Cora with adjacency matrix $\mathbf{A}_{\mathcal{S}}$ as visualized in (a). (b)–(i) are the $K$ outputs from the *edge partitioner*. Brighter color represents larger edge weights. For a clearer visualization effect, we enlarge the bright spots in (b)–(i) by 9 times.

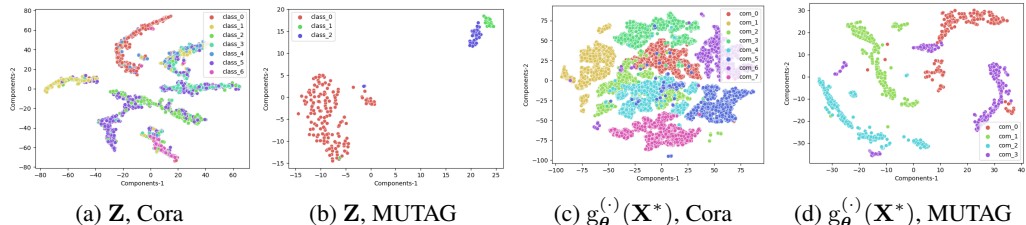

(a) $\mathbf{Z}$, Cora (b) $\mathbf{Z}$, MUTAG (c) $g_{\boldsymbol{\theta}}^{(\cdot)}(\mathbf{X}^*)$, Cora (d) $g_{\boldsymbol{\theta}}^{(\cdot)}(\mathbf{X}^*)$, MUTAG

Figure 4: t-SNE visualization of $\mathbf{Z}$ obtained from *unsupervised pretrain* on (a) Cora and (b) MUTAG, as well as community-specific node embeddings obtained during *supervised finetune* on (c) Cora and (d) MUTAG.

of assigned nodes, then sort the nodes within bucket $k$, $k \in \{1, \ldots, K\}$, by descending value of $\mu_{u,k}$. The $\mathbf{Z}$ used for sorting is sampled at the end of training.

The community structures that VEPM detects could be found in Figure 3a as the blocks on the main diagonal. The fact that most of the bright spots (*i.e.*, edges) are located in these on-diagonal blocks reflects dense node connections internal to each community. Sparsely distributed off-diagonal spots provide evidence of overlapping membership between the communities, which is also taken into account by our graph generative model. Figures 3b to 3i show that the *edge partitioner* has done a sound job in separating latent communities. Due to the limited size of the subgraph, some small metacommunities may not contain any large-weighted edges between the nodes in this subgraph, which accounts for the appeared emptiness in Figures 3h and 3i. Such kind of artifact disappears when we replace the subgraph by the full graph of Cora, as visualized in Figures 6 to 14 in Appendix G.

**Visualizing latent representations.** The previous visualization experiment verifies the following propositions: (i) the identified hidden structures are latent communities, and (ii) the partitioned graphs are different from each other. We now study how they benefit representation learning. Cora and MUTAG are selected as the representatives for node and graph classification benchmarks. For the MUTAG dataset, we remove graphs that contain node categories with less than 5 instances (less than 10 from a totality of 188) and randomly sample 10 graphs for the visualization experiments.

We first visualize $\mathbf{Z}$ obtained at the end of the *unsupervised pretrain* stage (Figures 4a and 4b), where the node-community affiliation vectors are projected to 2-D space via t-SNE [42]. Proposition (i) ensures that the node information carried by $\mathbf{Z}$ is about communities. We color code the scatters by node labels, both Figures 4a and 4b exhibit a strong correlation between the spatial clusters and colors, which indicates that even without label supervision, the node information provided with $\mathbf{Z}$ has discriminative power on classifying nodes.

We then visualize the community-specific node emebddings obtained at the end of the *supervised finetune* stage (Figures 4c and 4d). Similarly, t-SNE is adopted to reduce the dimensionality of obtained node embeddings. This time we color code the scatters by the latent metacommunity they correspond to. Both Figures 4c and 4d exhibit clear boundaries between the colored clusters, which show that the model is able to extract different information from multiple communities, which enhances the overall expressiveness of learned node or graph representations and potentially leads to better model performance on the downstream tasks.

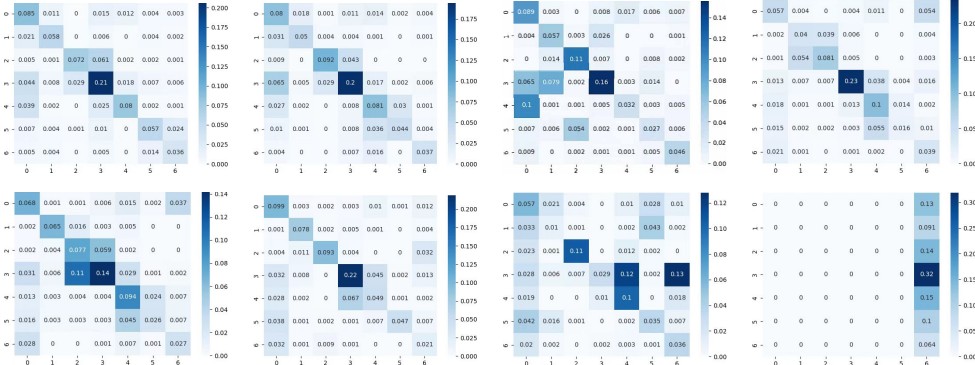

Figure 5: The 10-fold cross-validation results of using node representations learned from each of the eight communities for node classification, expressed in the format of normalized confusion matrices. The cell at the intersection of the $i$th row and $j$th column is the percentage of (mis)classifications that predict the label of an $i$th class node as $j$. A darker shade indicates a higher (mis)classification rate. Please note that the same color for different confusion matrices may correspond to different values.

## 5   Exploratory Studies

**Task-relevant communities.** We assess the relevance of the learned communities to the task from an experiment on the Cora dataset. In order to see the statistical dependency between communities and the task, we first hard-assign each node to the community of the highest affiliation strength, then we compute the normalized mutual information (NMI) between the hard-partitioned communities and node labels. NMI is a score between 0 and 1, with higher values indicating stronger statistical dependencies between two random variables. In the above experiment, we obtain an NMI of $0.316$ when the communities are learned solely by the *unsupervised pretrain*, such value increases to $0.322$ after the *supervised finetune*. The results of this experiment show that the initial communities we obtained from the *unsupervised pretrain* are meaningful to the task, and the relevance between the inferred communities and the task would be enhanced by the *supervised finetune*.

**Diverse community-specific information**. To see the effect of the information provided by different communities, we first train VEPM until convergence, then use the node representations learned by each community GNN to train an SVM and visualize the averaged 10-fold cross-validation results by a confusion matrix. With the results shown in Figure 5, we can further conclude that most of the communities can provide task-related information, as we can observe from Figure 5 that the diagonal cells in most of the confusion matrices have darker shades than off-diagonal cells. Beyond that, the information extracted from different communities is complementary to each other. For example, in the second column of Figure 5, information from the upper community can separate classes 3 and 4 well, but would mix classes 4 and 5; information from the lower community, on the other hand, can separate classes 4 and 5 but works worse on separating classes 3 and 4.

**The working mechanism of VEPM.** In summary, the communities learned by VEPM have the following properties: (i) the communities are relevant to the task; (ii) the information provided by different communities is complementary, so the overall amount of information for the task is accumulated over communities. Both properties are intuitively helpful for the downstream task.

## 6   Conclusion

Moving beyond treating the graph adjacency matrix as given, we develop variational edge partition models (VEPMs) to extract overlapping node communities and perform community-specific node feature aggregations. Specifically, we first utilize a GNN-based inference network to obtain node-community affiliation strengths, with which we augment node attributes and partition the edges according to the intensities of node interactions with respect to each community. We learn GNN-based node embeddings for each community by aggregating node features with the corresponding partitioned graph, and aggregate all community-specific node embeddings for the downstream tasks. Extensive qualitative and quantitative experiments on both node-level and graph-level classification

tasks are performed to illustrate the working mechanism and demonstrate the efficacy of VEPMs in supervised graph representation learning.

## Acknowledgments

Y. He and M. Zhou acknowledge the support of NSF IIS 1812699 and 2212418, and the Texas Advanced Computing Center (TACC) for providing HPC resources that have contributed to the research results reported within this paper. Besides, this work was supported in part by the National Natural Science Foundation of China under Grant U21B2006; in part by Shaanxi Youth Innovation Team Project; in part by the 111 Project under Grant B18039; in part by the Fundamental Research Funds for the Central Universities QTZX22160.

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
