# OpenReview forum: "A Variational Edge Partition Model for Supervised Graph Representation Learning"
_NeurIPS.cc/2022/Conference — NeurIPS 2022 Accept_

### Official Review · Reviewer_JKdP · 2022-07-08

**Rating:** 6
**Confidence:** 4
**Soundness:** 2 fair
**Presentation:** 3 good
**Contribution:** 3 good

**Summary:**

This works aim at modeling the community structure of graphs via the latent variable of the generative model, that captures which nodes are assigned to which communities and then leverages such information on message passing of GNNs, all of which are trainable in an end-to-end fashion with a variational inference framework. In particular,
* The authors model the community structure in the latent variable of the graph generative process, where they first calculate the affinity scores of nodes to the community and then use that scores for weighting edges during the aggregation of neighboring nodes in GNNs.
* The authors use different GNNs for different communities and then merge representations from them to make a prediction for downstream tasks.
* The authors formulate the proposed community-based generative model over a variational inference framework, making the proposed VEPM trainable.
* The authors verify the performance of the proposed VEPM on node and graph classification tasks and also conduct analyses to see its efficacy.

**Questions:**

### Major Questions and Suggestions
* Firstly, see the weaknesses above.
* As shown in Table 5, if the proposed VEPM including inference and generative networks is trained jointly without pre-training, it totally fails on downstream tasks when comparing its performance to other baselines. More explanations are needed for this particular behavior.
* I am wondering does the captured community by the proposed VEPM follow the community pattern extracted from existing community detection methods, for example, the METIS graph partitioning algorithm.


### Minor Questions and Suggestions
* Explaining how the generated edge weights are used in GNNs may be necessary. The form of a GNN layer in equation (1) lacks an expression of edges during neighborhood aggregation, which might be confusing for readers who are not familiar with GNNs.
* The claims on "state-of-the-art graph classification performance" should be tone-downed. The compared works are mostly before 2021, and there are lots of works achieving remarkable performances [2, 5].
* More analysis of the temperature scaling value in equation (4) would be valuable. I would like to see the chaining communities of the proposed VEPM by varying the temperature value (i.e., from soft to hard community assignments and their performances).
* The visualization in Figure 3 is extremely hard to see when I see this in a printed version. I would like to suggest authors visualize lesser nodes (e.g., 50 instead of the current 200).


---

[5] MaximumEntropy Weighted Independent Set Pooling for Graph Neural Networks. arXiv 2021.

**Limitations:**

The authors discuss the limitations and potential societal impact of their work in Section F of the supplementary file.

**Strengths And Weaknesses:**

### Strengths
* This work models the community structure into the latent variable of the graph generative process, which is reasonable, convincing, and interesting. I like the core idea that, the method captures community structures based on edge weights calculated from affinity scores of nodes to communities, which are then used for aggregating neighborhood features in GNNs.
* This paper is well-structured and easy to follow.


### Weaknesses
* There is relevant work [1, 2] that considers latent community (cluster) structures of nodes, which should be discussed.
* There is relevant work [3] that breaks down the entire training process into unsupervised training for capturing graph (edge) structures and supervised training for performing on downstream tasks, which is similar to this work's training process in Lines 182-192.
* The performance improvements are not significant. Specifically, in Table 2 and 3, the proposed VEPM has huge variances, and, considering the variance, the results are not significant against baselines. Also, the powerful FactorGCN model is only compared on two datasets, namely IMDB-B and MUTAG among eight different datasets, on which the FactorGCN may outperform the proposed VEPM.
* The authors should evaluate on recent benchmark datasets, namely OGB [4]. The used datasets, namely Cora, Citeseer, and Pubmed for node classification; TU datasets for graph classification, are relatively easy and somewhat outdated.
* The analyses on the MUTAG dataset may be problematic. This is a very small dataset only having less than 200 graphs, compared to other datasets (e.g., the PROTEINS dataset has more than 1,000 graphs), and also known as showing high variance due to its size. Thus, I suggest authors analyze their model with larger datasets.

---

[1] StructPool: Structured Graph Pooling via Conditional Random Fields. ICLR 2020.

[2] Accurate Learning of Graph Representations with Graph Multiset Pooling. ICLR 2021.

[3] Data Augmentation for Graph Neural Networks. AAAI 2021.

[4] Open Graph Benchmark: Datasets for Machine Learning on Graphs. NeurIPS 2021.

---

> ### Author Response · Authors · 2022-08-02
> **Response to reviewer JKdP**
>
> Thank you for providing us with potential related work and constructive suggestions about the experiments. With regard to the weaknesses you have pointed out and some suggestions, **we have made the following updates in the revised manuscript**:
>
> - We have referred to relevant work [1–3] in Section 2 (the first paragraph) and relevant work [4] in Section 3.4, and discussed their relation with VEPM.
> - We have updated the formulation of GNN and specified how edge weights are used in VEPM. This part of content is in Section 2 (the first paragraph) in the revised manuscript.
> - We have added more results of FactorGCN and GAT in Table 2 (graph classification). The source code we used for this experiment is from [FactorGCN’s official github repo](https://github.com/ihollywhy/FactorGCN.PyTorch).
> - We have enlarged the bright spots in Figure 3 for a clearer visualization effect.
> - We have conducted an ablation study on he effect of different selections of $\tau$, the results are added to Appendix F of the revised submission.
>
> Next, let us clarify your concerns.
>
> 1. *About benchmark and baseline selection.*
>
> - First, we agree with you that evaluations on more recent and larger datasets would enhance the significance of the proposed method. However, ogb typically involves a totally separate set of baselines that focus more on scalability and less on performance. We have been actively extending VEPM’s empirical evaluations to ogb, focusing on scaling our algorithm to extremely large graphs. We consider that extension as a separate work by itself and hence decide not to include it into this paper that is focused on developing the VEPM model and inference algorithm.
>
> - We choose these baselines (GATs, DisenGCN, FactorGCN, R-GCN, etc.) because they improve GNNs by enabling them to deal with multiple (hidden) relations. Since GNN is a fundamental model that could be applied to many tasks on graph-structured data, the improvement on GNNs would benefit more than one type of tasks. The graph pooling methods, competitive as they are, are exclusively for graph-level tasks. In this work, we mainly focus on making improvements on the GNN part, and thus the concept could be illustrated with simpler graph pooling modules. We consider combining the modeling power of VEPM and some powerful graph pooling methods as a separate work that is worth future investigation.
>
> - The graph datasets tested in the paper have been widely used for benchmarking purposes and hence allow us to make comparsion to a rich set of previously proposed algorithms readily available (though we still need to run the author-published code by ourselves on some datasets to fill in the missing baseline results). We consider these benchmark datasets as sufficient for the purpose of validating the merits of our VEPM model and inferene algorithm. Going beyond them is valuable, but given our limited computing resource, we decided to place higher priority on making a comprehenssive comparison on these common benchmarks and providing informative ablation studies.
>
>
> - *Improvements are not significant*.
>
>   Due to large variations across different validation folds, we agree for each individual dataset the improvement might not appear that significant (not only to us, this observation applies to almost all previous state-of-the-art methods). However, we'd like to note that our VEPM consistently ranks either the best or the second best aross different datasets, which is unlikely to happen by random and hence can be used to verify its apealing performance.

---

> > ### Author Response · Authors · 2022-08-02
> > **Response continued**
> >
> > 2. *Why training from scratch is not good.*
> >
> >   It is a good point that is worth discussion. We find that VEPM is dependent on the quality of edge partition, which is also shown in the ablation study on different partition schemes in Appendix F (rebuttal revision). A relatively meaningful edge partition as the initial state for finetuning would facilitate information exchange between the model and the task, and thus is beneficial for optimization. Contrariwise, a less meaningful edge partition as the initial state would make it more difficult for the model to converge to an optimal state.
> >
> >   We believe a less meaningful edge partition is the cause of VEPM's inferior performance than baselines when trained from scratch. Technically, VEPM does not “fail” in this scenario because the classification accuracy is still better than random guess, a more strict assessment would be “somewhere between baseline and random guess.” Our explanation of this behavior is as follows. Considering an extreme case where edge partition is not relevant to the task, which would render community-specific node representations meaningless for the task, and by passing them to the *representation composer*, which is a layer-reduced GIN, it is plausible that a worse result would be obtained. In the meantime, since the original graph is used in the *representation composer*, which still carries some information about the task, the model performance is protected from a total failure. We will add the above discussion to further revision.
> >
> > 3. *The patterns of the detected communities.*
> >
> >   We summarize the patterns of communities detected by some well-established community detection / graph clustering / graph partition algorithms, and compare them with VEPM.
> >
> > | patterns | METIS [5] | Spectral Clusetring / Min-cut [6]| SBM [7] | MMSB [8] | VEPM |
> > | :--- | :----: | :---: | :---: | :---: | :---: |
> > | Equisized | YES | N/A | N/A | N/A | N/A |
> > | High (low) local edge density | YES | YES | YES | YES | YES |
> > | Overlapping membership | NO | NO | NO | YES | YES |
> > | Task-dependent | NO | NO | NO | NO | YES |
> >
> >   - Equisized: the size of communities are approximately the same.
> >   - High (low) local edge density: the intra-community connection rate is specified to be high or low.
> >   - Overlapping memebership: nodes are permitted to be affiliated with multiple communities simultaneously.
> >   - Task-dependent: the community membership is affected by a downstream task.
> >
> >
> > ### *We hope the revision in the paper and the above discussions clarify your concerns and will be taken into consideration for further evaluation of our paper*.
> >
> > #### References
> >
> > [1] Yuan, Hao, and Shuiwang Ji. "Structpool: Structured graph pooling via conditional random fields." *Proceedings of the 8th International Conference on Learning Representations*. 2020.
> >
> > [2] Baek, Jinheon, Minki Kang, and Sung Ju Hwang. "Accurate Learning of Graph Representations with Graph Multiset Pooling." *International Conference on Learning Representations*. 2021.
> >
> > [3] Nouranizadeh, Amirhossein, et al. "Maximum Entropy Weighted Independent Set Pooling for Graph Neural Networks." *arXiv preprint arXiv:2107.01410* (2021).
> >
> > [4] Zhao, Tong, et al. "Data augmentation for graph neural networks." *Proceedings of the AAAI Conference on Artificial Intelligence*. Vol. 35. No. 12. 2021.
> >
> > [5] Karypis, George, and Vipin Kumar. "A fast and high quality multilevel scheme for partitioning irregular graphs." *SIAM Journal on scientific Computing* 20.1 (1998): 359-392.
> >
> > [6] Dhillon, Inderjit S., Yuqiang Guan, and Brian Kulis. "Kernel k-means: spectral clustering and normalized cuts." *Proceedings of the tenth ACM SIGKDD international conference on Knowledge discovery and data mining*. 2004.
> >
> > [7] Holland, Paul W., Kathryn Blackmond Laskey, and Samuel Leinhardt. "Stochastic blockmodels: First steps." *Social networks* 5.2 (1983): 109-137.
> >
> > [8] Airoldi, Edo M., et al. "Mixed membership stochastic blockmodels." *Advances in neural information processing systems* 21 (2008).

---

> > > ### Comment · Reviewer_JKdP · 2022-08-07
> > > **Thank you for your responses, however, one concern still remains.**
> > >
> > > I sincerely appreciate the authors' detailed responses and their effort in addressing my concerns/suggestions. I have read the response comments and the revised version of the submitted manuscript, which I further comment on one by one as follows:
> > >
> > > Q1. About benchmark and baseline selection.
> > >
> > > Regarding this point, I don't agree with the authors' opinion that the used datasets are enough to verify the performance of the proposed VEPM against other baselines. As the authors pointed out, the used graph classification datasets have high variances across different folds due to their small size (e.g., MUTAG dataset has less than 20 instances for the test), and the used node classification datasets, namely Cora, CiteSeer, and PubMed, are currently not regarded as the good datasets to show the performance of several models. As there are many good alternative benchmarking datasets [A, B], why not try them for evaluation? By using those benchmark datasets, I think the weakness -- marginal performance improvement of the proposed VEPM -- would be perhaps resolved.
> > >
> > > [A] Benchmarking Graph Neural Networks. arXiv 2020.
> > >
> > > [B] Open Graph Benchmark: Datasets for Machine Learning on Graphs. NeurIPS 2020.
> > >
> > > Q2. Why training from scratch is not good.
> > >
> > > The discussions, which the authors made regarding the failure of joint training of inference and generative networks without pre-training, are convincing. I suggest authors carefully include them in the revision, as the discussions are, for me, highly valuable.
> > >
> > > Q3. The patterns of the detected communities.
> > >
> > > Thank you for providing analyses on detected communities across different community detection methods including the proposed VEPM. I think it is worthwhile to include the results and discussions in the main paper.
> > >
> > > To sum up, I still have a concern on Q1 (See Q1. above for details).

---

> > > > ### Author Response · Authors · 2022-08-09
> > > > **Thank you for the feedback**
> > > >
> > > > Thank you for the additional feedback. We will follow your suggestions to add the discussions on "why training from scratch is not good" and the analyses of the characteristics of the communities detected by VEPM as well as some well-established community detection algorithms to the paper.  We will carefully arrange the contents if we are permitted to have extra pages.
> > > >
> > > > To address your concerns on the need of additional benchmarks, we are working on evaluating VEPM and related baselines on more recent large datasets. We have just obtained initial node-classification results on WikiCS [1], which are consistent with our findings on the other benchmarks tested in the paper: using the same set of parameters for PubMed ($i.e.$, without cross-validating the parameters on WikiCS), we have achieved  $78.7\pm 0.6$ accuracy on the task, while the accuracies of GCN, GAT, and SuperGAT on the same task, as reported in Kim et al [2], are  $74.0 \pm 1.0$ ,  $77.6\pm 0.6$ , and  $77.9\pm 0.5$, respecitvely.
> > > >
> > > > #### References
> > > >
> > > > [1] Mernyei, Péter, and Cătălina Cangea. "Wiki-cs: A wikipedia-based benchmark for graph neural networks." *arXiv preprint arXiv:2007.02901* (2020).
> > > >
> > > > [2] Kim, Dongkwan, and Alice Oh. "How to Find Your Friendly Neighborhood: Graph Attention Design with Self-Supervision." *International Conference on Learning Representations*. 2021.

---

> > > > > ### Comment · Reviewer_JKdP · 2022-08-09
> > > > > **Thank you for your reply**
> > > > >
> > > > > Thank you for further engaging with me.
> > > > >
> > > > > Regarding my suggestions on 1) more explanations on results of training from scratch and 2) more analyses on community detections, I hope the authors clearly include them in their paper, in the next revision.
> > > > >
> > > > > Regarding my concern about more large and recent benchmark datasets, I am happy to see that the authors are currently working on them, and I hope the authors include other benchmark datasets in their next revision.
> > > > >
> > > > > To summarize, this is a good paper, and the main reason I leaned toward the rejection of this paper is due to insufficient evaluations. However, the authors are making an effort to address them, and, based on the initial results that the authors have obtained thus far, I am happy to increase my score from borderline reject to weak accept. If the authors satisfactorily reflect all the above discussions/comments in their paper, I can further increase my rating. Best of luck.

---

### Official Review · Reviewer_ZMJP · 2022-07-11

**Rating:** 7
**Confidence:** 3
**Soundness:** 4 excellent
**Presentation:** 3 good
**Contribution:** 3 good

**Summary:**

This work proposes a novel variational auto-encoder that learns to separate the graph into different communities, and then perform a graph learning task by aggregating over those separate inferred community subgraphs.

The model works by learning a variational distribution for a node affiliation matrix, which essentially says how affiliated a node is with each community. From that, edges are are sampled using a poisson distribution based on how well adjacent nodes' affiliations overlap. The edges are then partitioned using an edge partitioner. Once partitioned, each community graph goes through a separate GNN encoder, and then the representations are composed using another model. The overall training algorithm has three objective terms corresponding to the classification task, the reconstruction task, and the KL divergence regularization.

The model is evaluated on node and graph classification tasks against a suite of relevant baselines. In addition, multiple visualizations and ablations are performed.

**Questions:**

In no particular order:
(Some of these are just open questions/ablations that do not necessarily have to be performed)

1) In figure 1, should the concatenation of X and Phi be horizontally concatenated instead of vertically?

2) (ablation) What would happen if there was no edge partitioner and all edges were passed to each GNN? Or randomly partitioned?

3) (ablation) What if the composer was just a mean of the representations? Or weighted sum of the representations based on each nodes' affiliation scores? This would further tests the idea that the GNN community bank is doing a significant amount of work.

4) Are there any datasets with community labels that we could see if the learned communities align with?

**Limitations:**

Adequately addressed

**Strengths And Weaknesses:**

Originality:

++ Seems to be a novel model overall and involves interesting ideas to learn these community associations through an unsupervised process.

Quality:

++ Paper is a complete work, it is well evaluated. In addition, the authors perform some good ablation studies and visualizations.

-- I would like to see more ablations indicating where the model improvement comes from. How much does the edge partitioner matter? How about the composer? Would replacing these with super simple versions hurt performance? I think there are some questions potentially about where the gains in performance might be coming from

- The analysis of space complexity is incorrect. It shows the parameter space complexity, not the space complexity. For example, if a model requires an N x N adjacency matrix in memory, that is O(N^2) space complexity even if there are no parameters. (This might just be a clarity issue, the authors should specify that this is analysis of the number of parameters).

Clarity:

++ Paper is very well written, and is precise.

+ Figure 1 is very helpful

- Figure 1 could be improved by incorporating the different loss terms in the diagram.

-Should probably specify that the optimization in equation 6 is maximization.

Significance:

++ Model is definitely significant in that it improves the state of the art as well as introducing some new ideas.

-- Unclear what the scalability of the method is in practice. No large graphs were tested.

Grammar/Spelling:

line 116: disjuction -> disjunction
line 271 "amount of label" -> "amount of labelled data"

---

> ### Author Response · Authors · 2022-08-02
> **Response to reviewer ZMJP**
>
> We appreciate your affirmation and valuable suggestions. Here are our thoughts about your suggestions and concerns.
>
> 1. *Adding ablations on random and no partition,  and simpler representation composer*.
>
>   It is a very good point. We have done several experiments and found some interesting results. In our ablation study on edge partition, we used two partition schemes. One is evenly partitioning weights to all communities, yielding $\mathbf{A}^{(1)} = \mathbf{A}^{(2)} = \cdots =\mathbf{A}^{(K)}$, which is analogous to “no partition.” Another is sampling the unnormalized edge weights from $\mathrm{Uniform}(0,100)$ then normalizing them along the community dimension with a $\mathrm{softmax}$ function. Generally speaking, the performance of VEPM with even edge partition is comparable to that of GIN, our base model; when we substitute random partition for even partition, the model performance becomes worse than the base model. These ablation study results suggest that VEPM can benefit from enhancing the quality of edge partitions.
>
>   Next, we use the partition obtained from a regular VEPM to represent partitions that are meaningful for the task, and a random partition to represent “meaningless partitions.” For each partition scheme, we contrast the performance of VEPM between two designs of the  *representation composer*: (1) GNN-based (the current design) and (2) a fully-connected layer (a simpler design). When the edge partition is meaningful, *i.e.*, relevant to the task, the overall performance with a simpler *representation composer* is (consistently) slightly worse than the current design, but still better than most of the baselines. However, if the partition is irrelevant to the task, the GNN-based *representation composer* would be the last thing standing between the model and a failed task (because the graph structure used for neighborhood aggregation is meaningful for the task). So the current GNN-based *representation composer* protects the performance of the model in the worst-case scenario. **The results and discussions about these experiments are added to Appendix F of the revised manuscript.**
>
> 2. *Errors in the “space complexity.”*
>
>   Thank you for pointing out our misuse of the term “space complexity.” As you have already noticed, what we originally intended to analyze is the number of parameters. **We have updated the section for complexity analysis in Section 4.4 of the revised manuscript.**, in which we analyze the time, space complexity as well as the number of parameters in a more rigorous $N/M/K/F/L$  expression ($K$: \#metacommunities, $M$: \#edges, $L$: \#layers). As for the memory cost of $\mathbf{A}^{(1)}, \cdots, \mathbf{A}^{(K)}$, it would be $\mathcal{O}(KM)$ since we only compute the partition weights where there is an edge. Since the graphs that most GNNs deal with are sparse, we can assume that $N^2 \gg M$ hence the space overhead created by edge partition is less than $\mathcal{O}(N^2)$.
>
> 3. *Incorporating loss terms into Figure 1 & direction of concatenation*.
>
>   Thank you for the suggestion, the loss terms are incorporated in the diagram in the revised manuscript. We use transpose notation of $\mathbf{\Phi}$ and node embeddings in Figure 1 so the concatenations are along the vertical direction. We have explained the concatenation direction in our revised caption.
>
> 4. *Scalability of the method*.
>
>   With the subsampling-based acceleration algorithm introduced in Appendix E, we are able to reduce the time complexity of training VEPM from $\mathcal{O}(N^2)$ to $\mathcal{O}(M)$, which is comparable to other GNN-based methods. This acceleration method has been tested on PubMed, which has around 20k nodes. **Results and discussions of this experiment are in Appendix E**. We have been actively investigating VEPM with subsampling-based acceleration to scale it to even larger graphs, *e.g.*, graphs in ogbn whose node counts are in the order of millions. We consider that extension as a separate work by itself that is worth a separate publication.

---

> > ### Author Response · Authors · 2022-08-02
> > **Response continued**
> >
> > 5. *Comparing detected communities with groundtruth communities*.
> >
> >   It is not trivial to find a dataset that has the groundtruth community labels and node / graph labels at the same time. Even if such a dataset exists, it is not guaranteed that task performance is taken into consideration by the criterion that determines the partition of the communities. In an extreme case, if these groundtruth communities are irrelevant to the task, detecting them would even damage the task performance, which is not desirable for the task-oriented VEPM.
> >
> >   On the other hand, if the groundtruth communities are quite informative to the task, we can find that there would be statistical dependency between them and the communities detected by VEPM. For example, in the citation network Cora, the nodes with the same label are densely connected and nodes with different labels are loosely connected. Such “locally dense connection” is widely acknowledged as a pattern of community structure hence the node categories can be treated as community labels. In our experiment, we hard-assigned each node to one community and computed the normalized mutual information (NMI) between our communities and the communities defined with node labels. NMI is a score between 0 and 1, with higher values indicating stronger statistical dependencies between two random variables. The NMI we have obtained is 0.322, which points to a strong correlation between the communities detected by VEPM and the “groundtruth communities” defined by node labels.

---

### Official Review · Reviewer_YYdd · 2022-07-11

**Rating:** 7
**Confidence:** 3
**Soundness:** 3 good
**Presentation:** 3 good
**Contribution:** 3 good

**Summary:**

This paper proposed a variational edge partition method for supervised graph representation learning. The proposed method learns the community affiliation matrix $\mathbf{\Phi}$, generates $K$ partitioned graph representing different communities, and learn from all of when with two stages of GNNs. The idea of partitioning the graph into overlapping latent communities is very interesting, and extensive experimental results showed the effectiveness of the proposed method.

**Questions:**

Other than the weaknesses in the previous section, I also have two more questions:

1. I wonder why many numbers are missing from Table 2. I would suggest the authors to get them if possible.

2. In my understanding, the proposed method need to back-propagate through all $K$ adjacency matrices (as mentioned in w1). If this is correct, then the space complexity should be at least $O(N^2)$ instead of the given one in Section 4.4 (assuming $H << N$). If my understanding is wrong, please clarify.

**Strengths And Weaknesses:**

Strengths:

s1. This paper is clearly written and easy to follow.\
s2. The idea of edge partitioning with generative method is very interesting.\
s3. Extensive experiments validated the effectiveness of the proposed method.\
s4. The case study in Figures 3, 5-13 is shows that the proposed method successfully separates the community structures in the graph.

Weakness:

w1. It seems the method needs to back-propagate through all $K$ adjacency matrices $A^{(1)}, \dots, A^{(K)}$, which could be expensive on the space when the graph is large. \
w2. The complexity analysis in Section 4.4 is pretty vague. I wonder if the authors can analyze in exact terms such as $N/K/F$ instead of GCN/GIN.

---

> ### Author Response · Authors · 2022-08-02
> **Response to reviewer YYdd**
>
> Thank you for the suggestions on the complexity analysis and experiments. Here are some clarifications to your concerns.
>
> 1. *Complexity analysis*:
>
>   Thank you for pointing out the problem in space complexity analysis. The original paper misused the terminology “space complexity” to refer to “number of parameters,” which was why we did not include the partitioned graphs into the overhead. We have updated our complexity analysis in Section 4.4 of the revised manuscript. In revision, we have analyzed the computational and memory complexity as well as the number of parameters, in the $N/M/K/F/L$ notation (in our expression “$K$” is the number of metacommunities and “$L$” is the number of layers). **Please refer to Section 4.4 of our latest revision for details**.
>
>   It is noteworthy that although the method needs to backpropagate through $K$ adjacency matrices, as these matrices are sparse, the memory consumption for this part is not $\mathcal{O}(KN^2)$ but $\mathcal{O}(KM)$, where $M$ is the number of edges. This space overhead is comparable to multi-head GATs [1,2] and other factorization-based methods [3,4].
>
> 2. *Missing baseline results on some graph classification benchmarks*:
>
>   We agree that a comprehensive comparison with the baselines would enhance the credibility of our conclusions. In the original submission, we only include the results reported in the official paper of corresponding methods, unfortunately, the intersection of the tested benchmarks across these baseline models is quite small, which makes some rows in Table 2 seem empty. We are working on implementing these baseline models and testing them on the benchmarks that are missed out by their documentation. Except for R-GCN [5] and CompGCN [6] that require edge types (which are not provided by most of the datasets), we have evaluated GAT [1] and FactorGCN [7] on some relatively small benchmarks due to the limited duration of the rebuttal session. **The results are recorded in Table 2 in the revised submission**. The recently obtained results appear to be quite consistent with the original conclusions. We will keep working on filling these missing baseline values to prepare for the camera-ready version.
>
> #### References
>
> [1] Veličković, Petar, et al. "Graph Attention Networks." *International Conference on Learning Representations*. 2018.
>
> [2] Gao, Hongyang, and Shuiwang Ji. "Graph representation learning via hard and channel-wise attention networks." *Proceedings of the 25th ACM SIGKDD International Conference on Knowledge Discovery & Data Mining*. 2019.
>
> [3] Ma, Jianxin, et al. "Disentangled graph convolutional networks." *International conference on machine learning*. PMLR, 2019.
>
> [4] Yang, Yiding, et al. "Factorizable graph convolutional networks." *Advances in Neural Information Processing Systems* 33 (2020): 20286-20296.
>
> [5] Schlichtkrull, Michael, et al. "Modeling relational data with graph convolutional networks." *European semantic web conference*. Springer, Cham, 2018.
>
> [6] Vashishth, Shikhar, et al. "Composition-based Multi-Relational Graph Convolutional Networks." *International Conference on Learning Representations*. 2020.
>
> [7] Yang, Yiding, et al. "Factorizable graph convolutional networks." *Advances in Neural Information Processing Systems* 33 (2020): 20286-20296.

---

> > ### Comment · Reviewer_YYdd · 2022-08-08
> > **response to authors**
> >
> > I appreciate the authors' detailed response and update in the paper. The updated time/space complexity now looks much clearer. Nonetheless, I still encourage the authors to fill up Table 2 if possible.
> >
> > I've updated my score.

---

> > > ### Author Response · Authors · 2022-08-09
> > > **We appreciate your update of the score**
> > >
> > > Thank you very much for the additional feedback and updating the score. We are glad that the revision has addressed your concern about the complexity analysis. When making the revision, we found that by following the more rigorous $N/M/K/F/L$ expression, the complexity of VEPM is much more clearly demonstrated than our original manuscript. Thus we sincerely appreciate your constructive suggestions.
> > >
> > > As for the empirical results, we will not only continue filling the missing baseline results, but also try to add additional results on more recent large datasets. The initial results on WikiCS [1] for node classification are consistent with our current observations: using the same set of parameters for PubMed ($i.e.$, without cross-validating the parameters on WikiCS), we have achieved $78.7 \pm 0.6$ accuracy on the task, while the accuracies of GCN, GAT, and SuperGAT on the same task, as reported in Kim et al [2], are $74.0 \pm 1.0$, $77.6 \pm 0.6$, and $77.9 \pm 0.5$, respecitvely.
> > >
> > > #### References
> > >
> > > [1] Mernyei, Péter, and Cătălina Cangea. "Wiki-cs: A wikipedia-based benchmark for graph neural networks." *arXiv preprint arXiv:2007.02901* (2020).
> > >
> > > [2] Kim, Dongkwan, and Alice Oh. "How to Find Your Friendly Neighborhood: Graph Attention Design with Self-Supervision." *International Conference on Learning Representations*. 2021.

---

### Official Review · Reviewer_Sk5x · 2022-07-11

**Rating:** 7
**Confidence:** 2
**Soundness:** 4 excellent
**Presentation:** 4 excellent
**Contribution:** 4 excellent

**Summary:**

This paper introduces a method called "variational edge partition model" that is generative graph learning framework. VEPM views edges and labels as overlapping community structures. The authors formalize the training of VEPM, and perform experiments evaluating its performance.

**Questions:**

Q1. The location of the "Related Work" is a bit strange. I would actually move it before Variational Edge Partition Model. Is there a particular reason why you did it this way?

Q2. A bit of a theoretical analysis of why the current method works would make the paper more complete. Any plans to include some analysis?

Q3. Why not release the code to make your work reproducible and your claims stronger?

**Strengths And Weaknesses:**

Originality: The work has high originality as it views the edges of GNNs very differently compared to existing K-hop GNN idea.
Quality: The work is of quality.
Clarity: The paper is clearly written. I have pointed out a few minor points below though.
Significance. The contribution seems significant as I have indicated above for my point about the work's originality.

Minor points:
- Line 58, put a comma after "From these scores"
- Line 101, end the sentence at "impractical".
- Line 171, typo on "community"

---

> ### Author Response · Authors · 2022-08-02
> **Response to reviewer Sk5x**
>
> We appreciate your positive feedback, here are the clarifications to your concerns.
>
> > The location of the "Related Work" is a bit strange. I would actually move it before the Variational Edge Partition Model. Is there a particular reason why you did it this way?
>
> Our original thought was to first show the details of VEPM in the hope that the similarities and differences between our model and the related work might be more evident. We have now adjusted the presentation as you suggested.
>
> > A bit of a theoretical analysis of why the current method works would make the paper more complete. Any plans to include some analysis?
>
> Thank you for your consideration for the completeness of our paper. **We have added related discussion in Appendix G**. Since it is hard to find mathematically strict theories which quantify how much the community information extracted by VEPM could benefit the task, we aim to find some heuristic insights via exploratory studies. Through experiments we find that (1) the detected communities are often related to the task labels; (2) the information extracted from different communities are complementary to each other, so the overall amount of information for the task is accumulated over communities. Both properties are intuitively helpful for the downstream task.
>
> > Why not release the code to make your work reproducible and your claims stronger?
>
> The code to reproduce the experimental results was submitted as part of the supplementary material. We will release it to Github for public access soon after the acceptance of the paper.

---

> > ### Comment · Reviewer_Sk5x · 2022-08-09
> > **Response to Authors**
> >
> > Thanks for your detailed and informative response, and apologies for not realizing the submitted code in the supplementary material!
> >
> > I will keep my score as it is.

---

> > > ### Author Response · Authors · 2022-08-09
> > > **Thank you for the review**
> > >
> > > Thank you for your affirmation of VEPM and the constructive suggestions about the completeness of the paper. The discussions about how community information benefits VEPM is quite relevant to our topic, we appreciate your insightful question.

---

### Author Response · Authors · 2022-08-02
**General response to reviewers and paper revision summary**

We appreciate all reviewers for their thoughtful feedback and constructive suggestions. We are glad that the core idea to learn community-specific representations via neighborhood aggregation with partitioned weighted edges is considered as novel (Reviewers Sk5x, ZMJP) and interesting (Reviewers YYdd, ZMJP, JKdP). And we are pleased to see that the contributions of VEPM are generally acknowledged. We have revised the paper regarding the concerns expressed by reviewers:


1. We have rearranged the presentation order. As suggested by Reviewer Sk5x, we have relocated the related work in a new section called “Preliminaries and Related Work” between the introduction section and the model section. We have also added some background knowledge of GNNs in this part, in which we not only show the general propagation function of a GNN layer, but also specify how the edges weights are used in the propagation function in our implementation, following the suggestion by Reviewer JKdP. We have also added some discussions about graph pooling. Besides, we have revised the description of these related work to highlight their similarities and differences with VEPM.

2. We have incorporated the correponding loss terms into Figure 1, as suggested by Reviewer ZMJP, and edited the caption.

3. **We have run additional experiements to fill in the missing results of baselines in Table 2 as much as we can**. Please see our Response 2 to Reviewer YYdd for more details.

4. We have enlarged bright spots in Figure 3(b) – 3(i) for a clearer visualization, so the size of each bright spot is comparable to adjacency matrices with fewer nodes while keeping as many visible community blocks as the adjacency matrix with 200 nodes. Caption is edited accordingly.

5. **We have re-evaluated the complexity of VEPM**, in which we used the “$N/M/K/F/L$” notation to express the time complexity, space complexity, and parameter number. We agree with Reviewer YYdd that this would make the presentation clearer. **Please refer to Section 4.4 for our revised complexity analysis**.

6. **We have extended the content of ablation studies in Appendix F**. The new studies focus on analyzing (1) the effect of edge partition, as suggested by Reviewer ZMJP; (2) the effect of alternative designs of the *representation composer*, as suggested by Reviewer ZMJP.

7. **We have added an exploratory study on the working mechanism of VEPM in Appendix G**, as suggested by Reviewer Sk5x.

Besides, **we are working on the ablation study about the effect of $\tau$**, as suggested by Reviewer JKdP, and will incoporate the results upon completion.

All the above revisions are marked in blue text color to make them easier to spot.

---

> ### Author Response · Authors · 2022-08-05
> **Response continued**
>
> Update 08/04/2022
>
> -- **We have added the ablation study on $\tau$ into Appendix F of the revised submission**, as promised in the previous response.

---

### Meta-Review · Area_Chair_HdzX · 2022-08-27

**Recommendation:** Accept
**Confidence:** Certain

**Metareview:**

This paper has novel ideas and good experimentations, and is well written. In particular, the novelty is a highlight in this paper, as compared with many existing k-hop ideas. Overall, it is a good addition to the general GNN literature. There was some initial disagreement on the evaluation of the method, which was later addressed by the authors' detailed response.

**Award:**

No

---

### Decision · Program_Chairs · 2022-09-14

Accept